# Do My Reactions Outweigh My Actions? The Relation between Reactive and Proactive Aggression with Peer Acceptance in Preschoolers

**DOI:** 10.3390/children10091532

**Published:** 2023-09-09

**Authors:** Brenda M. S. da Silva, Guida Veiga, Carolien Rieffe, Hinke M. Endedijk, Berna Güroğlu

**Affiliations:** 1Department of Educational and Developmental Psychology, Institute of Psychology, Leiden University, 2333 AK Leiden, The Netherlands; crieffe@fsw.leidenuniv.nl (C.R.); bguroglu@fsw.leidenuniv.nl (B.G.); 2Departamento de Desporto e Saúde, Escola de Saúde e Desenvolvimento Humano, Universidade de Évora, 7004-516 Évora, Portugal; gveiga@uevora.pt; 3Comprehensive Health Research Centre (CHRC), Universidade de Évora, 7004-516 Évora, Portugal; 4Department of Human Media Interaction, Faculty of Electrical Engineering, Mathematics and Computer Science, University of Twente, 7522 NB Enschede, The Netherlands; 5Department of Psychology and Human Development, Institute of Education, University College London, London WC1E 6BT, UK; 6Department of Educational Sciences, Institute of Education and Child Studies, Leiden University, 2333 AK Leiden, The Netherlands; h.m.endedijk@fsw.leidenuniv.nl

**Keywords:** reactive aggression, proactive aggression, peer acceptance, peer relations

## Abstract

Aggressive behaviors negatively impact peer relations starting from an early age. However, not all aggressive acts have the same underlying motivations. Reactive aggression arises as a response to an antecedent behavior of someone else, whereas proactive aggression is initiated by the aggressor and is instrumental. In this study, we aim to understand the relation between reactive and proactive aggression and peer acceptance in preschoolers. Parents of 110 children aged between 3 and 6 years old rated their children’s manifestation of reactive and proactive aggressive behaviors. To assess the children’s peer acceptance score within their class, they completed a paired comparisons task. The outcomes confirmed that reactive aggression in particular is negatively related to peer acceptance at the preschool age. Our results provide insights for the needs and directions of future research and interventions.

## 1. Introduction

All children develop emotionally and socially through social interactions with their peers. Positive peer interactions at school create a sense of belonging and are related to better mental health [1,2]. Yet opportunities to play are affected by how much a child is accepted by other children in the peer group, i.e., peer acceptance [3,4]. In the context of preschool, peer acceptance reflects the degree to which a child is liked by peers within their peer group [5]. Thus, accepted children are those who obtain more liking nominations from peers, which is associated with more shared affect and companionship [5]. Aggressive behaviors negatively affect peer acceptance. Studies focused on preschoolers’ aggression show that higher levels of aggression are related to lower peer acceptance, and more victimization, rejection, and conflicts within the peer group [6,7,8,9]. Furthermore, studies with elementary-school-aged children and adolescents that distinguish between the motives of aggression—reactive versus proactive—show that these two forms of aggression are differentially related to peer acceptance and peer relationships [10,11,12]. However, even though reactive and proactive aggression can already be distinguished in preschoolers [13,14], the relation between these two forms of aggression and peer acceptance is yet to be studied in this age group, which will be the focus of the current study.

Aggressive behaviors can be observed in toddlers after their first year of life, and tends to be mostly physical, manifested through behaviors such as biting, kicking, and hitting [15,16]. Albeit undesirable, these behaviors are not yet viewed as defiant, as they reflect the immaturity in regulating and communicating the toddler‘s own negative feelings [16,17,18]. These behaviors tend to increase until children are 3 to 4 years of age, yet decrease rapidly after entering a peer group [19,20], which can be explained by the social information processing (SIP) model developed by Crick and Dodge [21]. The SIP model suggests that children’s responses to new social situations are influenced by their daily social experiences, and that behavioral responses arise from the way that children process social cues based on six cognitive steps: (i) encoding of cues, (ii) interpretation of cues, (iii) clarification of goals, (iv) response construction, (v) response decision, and (vi) behavioral enactment [21,22]. Previous research has shown that children who act aggressively have atypical processing of the aforementioned steps, encoding more hostile cues (step i), interpreting situations as more hostile (step ii), more often opting for revenge or personal gain as a goal (step iii), constructing and deciding on aggressive responses more often (steps iv and v), and evaluating the impact of their aggressive behaviors towards others less negatively (see [23] for a review). Although not part of the original SIP model, emotion processing is understood as an important and interrelated additional element. For example, biased encoding and interpretation (steps i and ii) are related to impaired emotional understanding [24,25] and aggressive response construction. Decision and enactment are related to impaired emotion regulation [24,26]. As children grow older, their emotion processing skills are expected to improve and they become more skillful in navigating through the aforementioned steps. However, when these emotion-processing skills (e.g., emotional regulation; emotional understanding) are impaired, children are more prone to maintain these aggressive behaviors [7,27,28,29]. Note that biases in different steps result in different types of aggression. For example, biases in interpretation (step ii) are associated with reactive aggression, whereas biases in response decision (step v) are associated with proactive aggression [24,30].

Reactive aggression is a defensive and impulsive aggressive response to an antecedent behavior or provocation. In light of the SIP model, reactive aggressive children have an underlying tendency to encode more hostile cues (step i) and interpret the situations (step ii) as more hostile than their peers [24,30]. These biases are related to impaired emotional understanding and emotional regulation [24,30]. Thus, reactive aggressive children show an impaired ability to understand their own and other’s emotions, causing them to encode and interpret the situations as hostile. The difficulties in regulating their emotions can lead to quick escalation to an aggressive response (physical or verbal) [24,25,30]. More recently, Verhoef and colleagues [22] have suggested that over time, children who react aggressively seem to automatize these reactive aggressive behaviors as an immediate response to perceived negative and hostile behaviors by their peers, which can lead to social problems [31,32]. Indeed, studies with school-aged children and adolescents consistently show that more reactive aggressive behaviors are related to a lower peer acceptance within the peer group [10,12]. A longitudinal study has also shown that reactive aggressive behaviors during adolescence contributed to the prediction of a lower peer acceptance 18 months later [33]. Although—to the best of our knowledge—no study has addressed the relation between reactive aggression and peer acceptance for preschoolers, other constructs related to social functioning and aggression have been studied. For example, preschoolers who tend to be reactively aggressive are more often rejected within the peer group [13].

Proactive aggression is intentional and unprovoked aggressive behavior that is used to achieve a personal goal, hence it is an instrumental form of aggression [34]. In light of the SIP model, proactive aggressive children have a tendency to decide on the response (step v) that allows them to obtain personal gain, causing conscious harm to others [24,30]. Impaired empathy and emotional awareness can cause these children to devalue the impact of their actions on others’ well-being [24,35]. The few studies that have explored peer acceptance and proactive aggression in older children have highlighted that proactive aggression is related to a lower acceptance by the peer group [36,37]. However, proactive aggressive behaviors in adolescents did not predict later peer acceptance [33]. A longitudinal study that included reactive and proactive aggressive elementary-school-aged boys also showed that reactive aggressive boys have increased difficulties with peer interactions and a lower peer acceptance in comparison to proactive aggressive boys [38]. The relation between proactive aggression and peer acceptance is yet to be studied in preschoolers, although some studies have examined the relation between proactive aggression and other constructs related to social interactions in preschoolers. A study conducted by Evans and colleagues [13] indicated that displays of proactive aggression were related to teachers’ perceptions of the child being rejected/isolated from the peer group.

To the best of our knowledge, no study has focused on sex differences in the association between peer acceptance and preschoolers’ aggression. Furthermore, research on sex differences regarding the prevalence of reactive versus proactive aggression during preschool years has shown inconsistent findings, with some studies showing that boys were rated as more aggressive [10] and others studies showing that these behaviors are equally manifested by boys and girls [29,39].

## 2. Present Study

Peer acceptance is an important indicator of preschoolers’ social interactions and a predictor for later maladjustment. Understanding ‘if’ and ‘how’ each type of aggression relates to peer acceptance at this young age may provide better guidance for the development of early intervention to prevent social maladjustment and allow us to develop more specific and targeted interventions for young children. The first aim of this study was to examine the extent to which reactive and proactive aggression are related to peer acceptance in preschoolers. Considering that peer rejection is related to both types of aggression in this age group [13], we expected a negative relation of both reactive and proactive aggression with peer acceptances in preschoolers. The second aim of the study was to examine which type of aggression (i.e., reactive versus proactive) is more strongly related to peer acceptance. Our assumptions for these specific research questions are supported by longitudinal studies that indicate that reactive aggression alone has been shown to predict a lower peer acceptance [33] and that reactive aggressive boys are less preferred within the peer group in comparison to proactive aggressive boys [38]. Considering these findings, we expected the relation between reactive aggression and peer acceptance to be stronger compared to the relation between proactive aggression and peer acceptance. Previous research works focused on sex differences regarding preschoolers reactive and proactive aggression are inconsistent [10,29,39]. Therefore, we did not formulate specific hypotheses regarding sex differences but explored possible sex differences.

## 3. Method

### 3.1. Participants and Procedure

A total of 110 children aged between 3 and 6 years old (53% boys; *M_age_* = 61.26 months, SD = 10.05) participated in the study. The participants were recruited directly at preschools in the area of Lisbon, center and south of Portugal. In total, nine classes, each one from a different preschool, participated in the study, with average class size of eleven children (SD = 4.03). This study forms part of a larger study on deaf and/or hard of hearing (DHH) and hearing children. Children from the DHH group were recruited from hospitals in the area of Lisbon, and their parents were asked which preschool their children attended. These preschools were contacted, informed about the purpose and planning of this larger study, and their participation was requested. For preschools that agreed to participate, all parents from participating classes were given information about the study and asked to provide written consent for their children to participate. Parents and teachers also filled out paper questionnaires for every participating child. In order to reliably calculate the peer acceptance scores for the current study, only classrooms with a participation rate of at least 80% were included.

In order to allow the children to become familiarized with each other, all assessments took place at least two months into the start of the school year. The researchers went to the preschools on two different days. On the first day, the examiner met the children, explained the study and asked them for their verbal consent to participate. On the second day, children performed the peer acceptance task in a quiet room with the examiner.

### 3.2. Ethical Considerations

Approval for the study was obtained from the Portuguese Committee of Data Protection and from the Ethical Committee of the Portuguese Directorate of Education. The study complies with the General Data Protection Regulation (GDPR-2016/679) of the European Union.

## 4. Variables and Materials

### 4.1. Peer Acceptance

Peer acceptance was obtained using a computerized assessment of paired comparisons [40]. On the first day of the assessment, an individual picture of each child from the class was taken and uploaded to one tablet. All pictures were numbered so that each child was assigned to a specific and random number on the classroom list. In a separate room, each child was shown the screen of the tablet. First, to introduce the task, each child was presented with pictures of a pair of toys (either a bike/ball, bike/train, or ball/train) on the screen and asked to touch the toy that he/she liked to play with the most. Once it was clear that the child had understood how this worked, the experimental task started.

During the experiment, pairs of photos of classmates were presented on the screen, and the experimenter asked “which one of these children do you like to play with the most?”. The child was again asked to indicate his/her acceptance regarding a pair of photos by touching the screen, and the procedure was repeated. Initially, the maximum number of pairs (with random matching of photos of all children in the classroom; no pairs were repeated, thus all pairs presented were unique combinations) that were to be shown to each child was set up to 80; however, after the first four data collection sessions, it was clear that the younger children were distracted and uneasy after 40/50 pairs of the experiment. Therefore, the maximum number of pairs presented was adjusted to 45 to ensure that all children could stay focused throughout the task. Comparisons between the average results when children performed the task after viewing 45 pairs and 80 pairs revealed no significant differences in the average peer acceptance score between groups (*t*(133) = −0.906, *p* = 0.183); therefore, we maintained all results in our analysis. For every participant, a peer acceptance score was calculated by dividing the number of times the participant was chosen by each specific classmate by the number of times that the participant was presented on the screen. To achieve a general peer acceptance score for each participant, the peer acceptance scores received from all classmates were averaged and standardized within each class.

### 4.2. Reactive and Proactive Aggression

Reactive aggression and proactive aggression scores were obtained using the Aggressive Behavior Rating developed by Dodge and Coie [34]. The questionnaire was administered to parents; who were asked to report on their child’s manifestation of reactive aggressive behaviors (3 items; i.e., “When teased, strikes back”, “Overreacts angrily to accidents”, and “Blames other children for the fights”); and proactive aggressive behaviors (3 items; i.e., “Threatens or hits other children”, “Makes other children turn against one child”, and “Uses physical strength to dominate other children”). Parents rated on a 5-point scale (0 = (almost) never, 1 = rarely, 2 = sometimes, 3 = often, 4 = (almost) always). To obtain the reactive and proactive aggression scores, the average of the items belonging to each scale was calculated per child. The scores obtained for each child were then standardized within each class (i.e., z-score calculation within classes). Due to the small number of items in the reactive and proactive aggression scales, inter-item correlations were considered as the measure for internal consistency [41]. In both scales, the mean inter-item correlations were within the ideal range (0.39 for reactive aggression; 0.32 for proactive aggression), confirming the coherence of the items in each scale and that each scale is attending to the specificity of each type of aggression. The Cronbach’s alphas of both scales were within the acceptable range (0.67 for proactive aggression; 0.60 for reactive aggression).

## 5. Statistical Analyses

Considering that our data involves sociometric assessment, prior to data analysis we standardized the scores of all variables, as suggested by Coie and colleagues [42]. As there were classrooms with a hierarchical structure (a couple of children who are aggressive and most of the class shows no aggression) or with a more egalitarian distribution (everyone is equally aggressive), this preliminary step allowed us to attend to the specificity of each class, by comparing children with their peers and not the overall sample.

To answer our first research question, bivariate correlations between the variables of our study, i.e., peer acceptance, reactive aggression, and proactive aggression, were conducted. To answer the second research question comparing the strength of the correlations between each type of aggression and peer acceptance, the Meng’s Z test [43] was used. Considering our hypotheses that the relation between peer acceptance and reactive aggression is stronger than that between peer acceptance and proactive aggression, we used one-tailed testing to access significance. The Meng’s Z-test in this study was conducted using R’s (version 4.2.1) cocor package (version 1.1-4) [44]. All of the other statistical analyses were performed with the IBM SPSS (version 28). We analyzed the mean differences between the sexes for all variables using an independent sample t-test. Additionally, we explored the differences in the strength of the relation between both types of aggression and peer acceptance by conducting the Meng’s Z-test for boys and girls separately. Possible sex differences in the relation between each type of aggression and peer acceptance were analyzed using the Fisher r-to-z transformations.

This study was registered prior to data analysis. The registration, and deviations from the original research plan, can be assessed through OSF.

## 6. Results

The range, mean, and standard deviations of the raw and standardized scores of peer acceptance, reactive, and proactive aggression are reported in Table 1. Our correlation analyses (see Table 2 and Figure 1) to examine the relations between our variables yielded a positive relation between reactive and proactive aggression. As expected, peer acceptance was negatively related to reactive aggression, yet unrelated to proactive aggression. Comparisons between the strength of the correlations for each type of aggression and peer acceptance showed that the correlations of reactive and proactive aggression with peer acceptance did not differ (*z* = −0.95; *p* = 0.82).

Exploratory analyses regarding sex differences showed that boys and girls did not differ in their peer acceptance (*t*(108) = −3.26, *p* = 0.87) but differed in regards to reactive (*t*(108) = 0.12, *p* = 0.001) and proactive aggression (*t*(99) = −0.3.04, *p* = 0.003). The positive relation between both types of aggression that was observed for the overall group was also found in both sexes. Furthermore, the relation between reactive aggression and peer acceptance that we observed for the overall group was not significant when calculated for boys and girls separately. Since no relation was found between both types of aggression and peer acceptance for girls and boys separately, the strengths of these correlations were not compared. In addition to this, the Fisher r-to-z value revealed no sex differences regarding the strength of correlation between reactive aggression and peer acceptance (*z* = −0.78; *p* = 0.22), and proactive aggression and peer acceptance (*z* = 0.04; *p* = 0.52).

## 7. Discussion

Aggressive behaviors can negatively affect peer relations during the preschool years [8,9,45]; however, different underlying mechanisms of aggressive behaviors might also affect peer relations differently. Consistent with previous studies in preschool children, the current study confirmed that, even at this age, aggressive behaviors can be distinguished regarding their reactive or proactive nature [13,14]. Although several aspects of peer relations (e.g., victimization, rejection) have previously been studied regarding both types of aggression [13,46], this is the first study to focus on peer acceptance. Based on parent reports and peer nominations, our findings show that reactive aggression is related to preschoolers’ peer acceptance, whilst proactive aggressive behaviors were rarely reported and unrelated to peer acceptance. Yet the strength of the correlations between each type of aggression and peer acceptance did not differ in our sample. Exploratory analyses highlighted that the negative relation between reactive aggression and peer acceptance was absent when considering boys and girls separately, and no sex differences appeared. These results were partially in line with our expectations and will be further discussed below.

Unlike outcomes based on studies in older children [36,37], the expected relation for proactive aggression with peer acceptance was absent in our study. Note that proactive behaviors showed a low incidence. In fact, only 46% of parents noted at least one incidence of proactive aggression in their children, compared to 87% for reactive aggression. Although all studies on proactive and reactive aggression during the preschool years show a much lower incidence of proactive aggression compared to reactive aggression, it might also be that preschoolers are not yet aware of the manipulative aspect that proactive aggression holds (e.g., helping them in gaining or maintaining status) [47,48]. In addition, parents may also be unlikely to see their own child as proactive aggressive, i.e., having manipulative or instrumental goals, which might thus result in the relatively low rates of reported proactive aggressive behaviors.

Our exploratory results regarding sex differences showed that boys had higher rates in both types of aggression compared to girls, which is in line with some previous research [10]. However, these sex differences were not found to have an impact on the relation of peer acceptance with each type of aggression. As girls’ aggressive behaviors tend to be more difficult to report [49], girls’ aggressive behavior is possibly underestimated. Typically, girls exhibit relational aggression which involves causing deliberate harm to others’ social relationships and the way that they are perceived by others [49]. Therefore, future studies focused on the relation between aggression and peer acceptance should use more comprehensive instruments to measure aggression, combining the type (physical versus relational), and the form (reactive versus proactive) when looking into sex differences in preschoolers.

The findings of this study shed light onto the importance of the early prevention of reactive aggressive behaviors, as they are already impacting social relationships starting as early as the preschool years. The current study has several strengths that should be highlighted.

Firstly, considering that peer acceptance is built upon the interactions between preschoolers, the use of computerized paired comparisons allowed us to rely on peers’ perspective regarding their preferences, rather than those reported by teachers or caregivers, who may not fully grasp how preferences are underlying actual play behaviors. Moreover, this computerized assessment is appropriate for young children, since the task only involves comparisons between two pictures of peers, without demands on complex cognitive or language skills.

Furthermore, our results provide guidance for intervention in the preschool setting that might aid children’s SIP. As previously mentioned, reactive aggression stems from issues in the encoding and interpreting of social situations (steps i and ii), that are interrelated with difficulties in emotional processing, namely emotional understanding and emotional regulation [24,30]. Therefore, it could be essential at this young age to opt for interventions that are focused on body awareness (i.e., understanding the body cues that signal an increased anger arousal) and self-regulation [22], in order to attend to the specific difficulties of emotional processing that reactive aggressive children are negatively impacting their ability to process social information properly. For example, body-oriented interventions have been shown to positively contribute to preschoolers’ emotional processing [50]. Within the scope of body-oriented interventions, interventions that combine play and relaxation aid children to promote their self-regulation skills [51,52]. Furthermore, recreating scenarios that are encountered in their daily lives, and reinforcing problem-solving skills, may allow teachers/practitioners to improve children’s ability to perceive situations (steps i and ii) in a less hostile way and contribute to the remainder of the SIP steps [22]. Therefore, promoting these interventions in the preschool setting might contribute to reducing the prevalence of reactive aggressive behavior in this age group, which future studies could further explore.

Finally, our findings create a baseline for future studies focusing on more complex theoretical models and approaches. For example, recent studies have confirmed that the teacher–child relationship mediates certain aspects of peer relationships [53,54,55], especially for children who have behavioral problems (see [56] for a review). Children spend most of their time under the care of their teachers, and their peer responses are modelled by mimicking how teachers respond to their peers [57]. When teachers expose these children to regulated models, children acquire these positive models as their own [57]. When conflicts arise, teachers can help children to navigate through the situation by promoting changes in children’s encoding and interpretation (steps i and ii), by helping children to evaluate the initial situation, how they felt about it, what their perception was, and why they reacted in a certain way [22,24,30]. As behavioral responses are largely influenced by the social repertoire that children have, while guiding them in re-evaluating their responses to peers, encouraging perspective taking and problem-solving skills that children can internalize can provide alternative ways to respond to future social conflicts, rather than automatizing reactive aggressive patterns [22]. Therefore, future studies should consider the role of the teachers in the peer acceptance of reactive/proactive aggressive children. The potential of these studies is not only to make teachers more aware of their importance, but it is also crucial to provide them with tools to facilitate relationships with children when challenges arise (e.g., [51,52]), so that dysfunctional patterns are not repeated in daily interactions between peers.

Although this study gave new insights into children’s aggressive behaviors and peer relations in the preschool years, the use of parent-only reports is a limitation. The present results should be taken with caution, as the strength of the correlations could have been influenced by a low power due to our sample size. Future studies should aim to include teachers’ perspectives to assess reactive and proactive aggression. A second limitation is related to the cross-sectional nature of this study that prevents us from drawing conclusions about the possible impact of each type of aggression on peer acceptance across the preschool years. Finally, future studies would largely benefit from also including observational measures (e.g., free play interactions) that provide additional valuable information regarding the manifestation of proactive/reactive aggressive behaviors towards peers in the preschoolers’ natural environment (e.g., on the playground).

In conclusion, although both types of aggression, reactive and proactive, showed a rather low prevalence at the preschool age, our outcomes also highlighted that reactive aggression is negatively associated with children’s peer relations already in preschool age. As a final note, we suggest multi-method and multi-informant measurements for a complete assessment of proactive and reactive aggression during preschool years, as well as conducting longitudinal and experimental studies to further improve our understanding of the links between various forms of aggressive behaviors and the development of peer relationships.

## Figures and Tables

**Figure 1 children-10-01532-f001:**
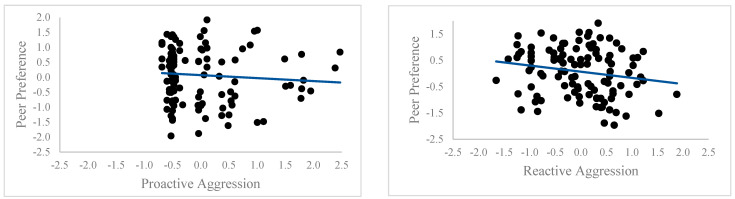
Scatter plots of the correlations between the standardized scores for reactive and proactive aggression, and peer preference for the overall group.

**Table 1 children-10-01532-t001:** Range, means, and standard deviations of peer preference, reactive and proactive aggression for the overall group, and separately by sex (raw score/standardized score—Z).

	Range Total (Raw Scores)		Mean Total (SD)		Mean Boys (SD)		Mean Girls (SD)	
	Min (% of Children)	Max (% of Children)	Raw	Z	Raw	Z	Raw	Z
Peer Preference	0.31 (1%)	0.71 (1%)	0.52 (0.09)	0.07 (0.92)	0.51 (0.09)	0.04 (0.93)	0.53 (0.09)	0.11 (0.93)
Reactive Aggression	0.00 (13%)	3 (1%)	1.28 (0.73)	0.00 (0.72)	1.47 (0.67)	0.20 (0.63)	1.07 (0.75)	−0.23 (0.75)
Proactive Aggression	0.00 (54%)	2.33 (2%)	0.34 (0.50)	0.00 (0.75)	0.49 (0.59)	0.19 (0.84)	0.17 (0.30)	−0.22 (0.55)

Note. The scale scores have been standardized within classrooms.

**Table 2 children-10-01532-t002:** Correlations between peer preference, reactive and proactive aggression for the overall group (boys/girls).

	Peer Preference	Reactive Aggression
Reactive Aggression	−0.18 * (−0.18 / −0.18)	-
Proactive Aggression	−0.08 (−0.13 / 0.02)	0.40 *** (0.38 **/0.33 **)

* *p* < 0.05, ** *p* < 0.01, *** *p* < 0.001.

## Data Availability

The data that supports the findings of this study are available from the corresponding author upon reasonable request.

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
