# Peer review of "Do My Reactions Outweigh My Actions? The Relation between Reactive and Proactive Aggression with Peer Acceptance in Preschoolers"

_children, 2023, doi:10.3390/children10091532_

Round 1
Reviewer 1 Report
Thank you for the opportunity to review this paper for Children. This is a correlational study with preschool-aged children focused on reactive aggression, proactive aggression, and peer acceptance. I have read the paper and found a lot of flaws and limitations with the study. I will note these below (in no particular order).
1. The sample size is extremely low for a correlational study with children. No power analysis was included to examine whether the study was adequately powered. I found an article that measured the correlation between aggression and peer acceptance in preschool samples (O'Toole, Monks, & Tsermentseli, 2016) and computed a power analysis based on their effect size (r = -.19) for a two-tailed test with alpha .05 and power = .80 and results showed that 212 participants are needed - which is roughly 2x the sample size the authors used. Thus, I do not trust the results due to unstable effect sizes based on low power.
2. The number of items for proactive and reactive aggression is 3 each. The authors could do a Cronbach's alpha for internal consistency rather than looking at the inter-item correlations. Moreover, why were parents given this measure instead of teachers? It seems odd that the peer acceptance happens at school but the aggression is at home where, presumably, the parents do not see the kid's school behavior. The former issue is easy to fix, but the latter one needs more justification.
3. The introduction needs to focus on the General Aggression Model or Social Information Processing Model, rather than Social Learning. Both models argue for the cognitive underpinnings that differentiate reactive from proactive aggression that can help guide the paper.
4. Based on the items provided to assess reactive and proactive aggression, I do not think that the authors adequately can differentiate the two forms of aggression with the items. Usually, the reactive items will include a prior statement to the item, such as "when someone makes me angry or upset, I will...." and the proactive adds a statement at the end, such as "I will often......to get what I want" (see Boxer et al., 2004 - Aggressive and Prosocial Behavior Questionnaire). Take one of the items provided...."Blames other children for the fights" could be reactive or proactive (and I am not convinced that "blaming" someone is an aggressive behavior). Moreover, how is "threatens or hits other children" only proactive? I only see one valid item that clearly maps onto reactive aggression: "when teased, strikes back". Other than that, the authors did not adequately measure the constructs intended.
5. Why was the aggression score standardized within each class? If the authors are worried about nested effects, then a multilevel model is needed; however, that will require a much higher sample size.
6. The statistical analyses are underwhelming. Moreover, a quick search of the past literature will show that the correlations between reactive aggression, proactive aggression, and peer acceptance have been done before (e.g., Marcus & Kramer, 2001; Poulin & Boivin, 2000; Price & Dodge, 1989). Therefore, I do not see how this study advances our understanding of this topic. A better approach would have been to examine whether peer acceptance and some other variable interacted to influence reactive, but not proactive, aggression. What that other variables could be is vast; however, the authors did not seem to measure any other variable that could fill this gap. Finally, neither the document nor the supplementary materials contain the tables or figure.
There were many typos and grammar issues. For instance, some in-text citations were repeated - one right after the other. The use of and vs. "&" in citations was inconsistent. Finally, periods were missing a the end of some sentences. Overall, a thorough editing is needed.
Reviewer 2 Report
I thank the authors for producing this interesting scientific paper. The research extends current knowledge, with a rigorous methodology and an explanation of the data in line with today's recognised hypotheses and theories.
Some suggestions may help to increase the quality of their work, considering that overall the work is satisfactory and in line with the quality of a scientific paper.
I suggest that the authors revise the introduction by trying to reduce, where possible, some elements that make the text seem redundant.
Furthermore, I suggest the authors define well what is meant by peer acceptance (there are various definitions and various measurement tools) tying the definition to the tools they use to measure the construct.
Furthermore, I think a discussion of gender differences should be included in the introduction.
In the discussion, which could be extended, I would give more space to theoretical interpretations of your results. Furthermore, I would update the literature as a whole and try to increase citations (see suggestions). Finally, I would emphasise the strengths of your study, before the weaknesses (the limitations). Finally, I would pay more attention to describing the practical implications of your study and, last but not least, emphasise the importance of investigating more complex theoretical models. For example, what is the role of the teacher? Could future studies consider the quality of the teacher-pupil relationship? And why should it? (see suggested references).
Suggested references
Prino, L. E., Longobardi, C., Fabris, M. A., & Settanni, M. (2023). Attachment behaviours toward teachers and social preference in preschool children. Early Education and Development, 34(4), 806-822.
Longobardi, C., Lin, S., & Fabris, M. A. (2022, April). Daytime sleepiness and prosocial behaviours in kindergarten: The mediating role of student-teacher relationship quality. In Frontiers in Education (Vol. 7, p. 710557). Frontiers.
Longobardi, C., Settanni, M., Lin, S., & Fabris, M. A. (2021). Student-teacher relationship quality and prosocial behaviour: The mediating role of academic achievement and a positive attitude towards school. British Journal of Educational Psychology, 91(2), 547-562.
Sette, S., Spinrad, T. L., & Baumgartner, E. (2013). Links among Italian preschoolers' socioemotional competence, teacher-child relationship quality, and peer acceptance. Early Education & Development, 24(6), 851-864.
Yue, X., & Zhang, Q. (2023). The association between peer rejection and aggression types: a meta-analysis. Child Abuse & Neglect, 135, 105974.
I suggest a check of inglish language by a native speakers.
Reviewer 3 Report
I read the Brief Report of B. M.S. da Silva at al entitled: “Do my reactions outweigh my actions? The relationship between reactive and proactive aggression with peer acceptance in preschoolers”. The subject of the manuscript is interesting, the introduction is adequate and the methodological approach seems correct. However, in the introduction a clarification of the term Proactive Aggression for preschoolers is needed.
My comments follow:
-In the method, specify how you selected the 9 kindergartens (are they representative?)
-In the method create a separate section for Ethical considerations, add if the study complies with General Data Protection Regulation (GDPR-2016/679) of the European Union.
-In the results ADD, tables 1 and 2 and image 1, which are absent.
- References should be numbered, according to the journal's instructions.
Round 2
Reviewer 1 Report
--
Reviewer 2 Report
Thank you for the revision. In my opinion, the paper can be published in the current form. Best regards.